# Highly Pathogenic Avian Influenza H5N8 Outbreak in Backyard Chickens in Serbia

**DOI:** 10.3390/ani13040700

**Published:** 2023-02-16

**Authors:** Biljana Djurdjević, Vladimir Polaček, Marko Pajić, Tamaš Petrović, Ivana Vučićević, Dejan Vidanović, Sanja Aleksić-Kovačević

**Affiliations:** 1Department of Epizootiology, Clinical diagnostics and DDD, Scientific Veterinary Institute “Novi Sad”, 21000 Novi Sad, Serbia; 2Department of Virology, Scientific Veterinary Institute “Novi Sad”, 21000 Novi Sad, Serbia; 3Department of Pathology, Faculty of Veterinary Medicine, University of Belgrade, 11080 Belgrade, Serbia; 4Veterinary Specialized Institute Kraljevo, 36000 Kraljevo, Serbia

**Keywords:** avian influenza, backyard, chickens, H5N8, H5N1, immunohistochemistry, pathology, Serbia

## Abstract

**Simple Summary:**

A novel reassortant highly pathogenic H5N8 avian influenza virus spread through migratory birds in many countries in Europe in 2016 and 2017. In November 2016, the virus was first detected in wild birds in Serbia, and shortly thereafter, it spread to backyard poultry in Vojvodina province, in the north of Serbia. In 2021/2022, new cases of avian influenza infection in wild birds and backyard poultry were reported in our country, and this time the H5N1 subtype dominated. The aim of this study is to describe the outbreaks of highly pathogenic avian influenza H5N8 and H5N1 in backyard chickens, including a detailed description of the pathological changes, clinical signs of the disease, and identification of the virus. Our results are consistent with those previously reported for poultry naturally infected with HPAIV H5N8 and H5N1 viruses and confirm that infection in chickens results in severe systemic disease with neurological manifestations.

**Abstract:**

In winter 2016/2017, the highly pathogenic avian influenza virus H5N8 was detected in backyard poultry in Serbia for the first time. The second HPAI outbreak case in backyard poultry was reported in 2022, caused by subtype H5N1. This is the first study that documents the laboratory identification and pathology associated with highly pathogenic avian influenza in poultry in Serbia during the first and second introduction waves. In both cases, the diagnosis was based on real-time reverse transcriptase PCR. The most common observed lesions included subepicardial hemorrhages, congestion and hemorrhages in the lungs, and petechial hemorrhages in coelomic and epicardial adipose tissue. Histologically, the observed lesions were mostly nonpurulent encephalitis accompanied by encephalomalacia, multifocal necrosis in the spleen, pancreas, and kidneys, pulmonary congestion, and myocardial and pulmonary hemorrhages. In H5N8-infected chickens, immunohistochemical examination revealed strong positive IHC staining in the brain and lungs. Following these outbreaks, strict control measures were implemented on farms and backyard holdings to prevent the occurrence and spread of the disease. Extensive surveillance of birds for avian influenza virus did not detect any additional cases in poultry. These outbreaks highlight the importance of a rapid detection and response system in order to quickly suppress outbreaks.

## 1. Introduction

All avian influenza (AI) viruses are segmented RNA viruses that belong to the *Influenzavirus A* genus of the *Orthomyxoviridae* family. Through history, influenza A viruses have been in the spotlight because they have caused multiple epidemics and pandemics in human and animal populations worldwide [1]. Influenza A viruses can be divided into two distinct groups based on their ability to cause a disease: the highly virulent viruses cause highly pathogenic avian influenza (HPAI), and all other viruses cause a much milder disease (Low Pathogenic Avian Influenza—LPAI) [2]. Highly pathogenic avian influenza viruses can cause a lethal disease in various bird species, which may result in flock mortality as high as 100%, leading to severe economic losses in the poultry industry. Due to their potential zoonotic nature, they also represent a continuous threat to public health [3].

Since 1996, numerous outbreaks of highly pathogenic avian influenza H5N1 infections in wild birds and poultry have occurred. After the first identification in Asia, H5N1 infections have spread to more than 60 countries in different parts of the world, including Europe, the Middle East and Africa [4,5]. Since then, H5N1 has become endemic in Southeast Asia in poultry, and co-circulation with other AI viruses led to the generation of novel reassortant H5 strains with neuraminidases (NAs) and internal gene assemblage from different prevalent subtypes of avian viruses [6]. Furthermore, this led to a significant accumulation of mutations in these viruses, resulting in the classification of H5 HA genes into more than 10 genetically distinct clades and subclades according to the phylogenetic analysis [7].

In 2010, an outbreak of novel reassortant HPAI strain H5N8 of clade 2.3.4.4 was reported for the first time in China; by 2014 this virus had caused multiple outbreaks in Asia (South Korea, Japan and China) [8], and soon after, in Europe (Germany, United Kingdom, The Netherlands) [9], Russia [10], and North America [11]. This was the first time since 2005 that a single strain of HPAI viruses had spread over such a vast geographical area [12]. During these outbreaks, two distinct groups of HPAI H5N8 viruses were identified: group A (Buan-like: A/Broilerduck/Korea/Buan2/2014) and group B (A/breederduck/Korea/Gochang1/2014, Gochang-like) [13]. Since the first epizootic in Europe in late 2014, the HPAI H5N8 has disseminated rapidly in other European countries. This intra- and intercontinental spreading of HPAI H5N8 was attributed to the wild bird migrations, and therefore their role in the epidemiology of this virus was pivotal [12,14]. From October to December 2016, 14 European countries, including neighboring Serbian countries Croatia, Hungary and Romania, reported numerous detections of H5N8 HPAI cases in poultry as well as in wild birds [15]. Many studies pointed out that a wide range of avian species, including wild and domestic waterfowl, domestic poultry, birds of prey, and even zoo birds, appeared to be susceptible to infection of the novel HPAI H5N8 [14,15,16].

In Serbia, the emergence of HPAI subtype H5N8 infection was first recorded in late November 2016. From November 2016 to March 2017, a considerable increase in mortality among mute swans was observed in different localities, mostly in the Province of Vojvodina, in the north of Serbia. During that period, mute swans were the most-affected wild bird species, with the highest percentage of mortality. Infected swans showed lethargy and the signs of central nervous system disturbance, such as incoordination, torticollis and other unusual positions of the head. Some of the results of these observations have already been published [17,18]. Besides that, a testing of other dead wild birds from the Province of Vojvodina was carried out and H5 subtype of AI virus was detected in two wild bird species, including one grey heron (*Ardea cinerea*) and one common buzzard (*Buteo Buteo*). Sequence analysis of these isolates was later identified as H5N8 (unpublished results). This first HPAI H5N8 outbreak in Serbia was immediately reported to the World Organization for Animal Health after its confirmation. In total, 24 outbreaks were reported, and most of them were located in the northern region of Serbia, in 8 different districts: South Bačka (11), North Bačka (1), Central Banat (3), West Bačka (1), Srem (2), and also in Braničevo district (3), Bor (1) and the city of Belgrade (2). Fortunately, during these outbreaks, none of the poultry farms was affected, so there were no losses for the poultry industry. Only four cases occurred in backyard poultry holdings in the South Bačka and Central Banat districts, where the poultry had not been sufficiently protected against contact with wild birds and where the level of biosecurity was low. 

After five years (during the 2021/2022 epidemiological year), new cases of AI were registered in our country. This time, the predominant subtype was H5N1, which was detected in four sporadic cases in wild birds (exclusively in mute swans) and in three isolated cases in backyard poultry. All HPAI cases were detected in the north of the country, in Vojvodina region, which is characterized by a large number of aquatic ecosystems which are a habitat for migratory bird species. No further spread of the disease to other parts of the country was detected, and not a single case was reported in commercial poultry. Compared to the previous Serbian HPAI outbreak, a lower mortality rate was observed in both wild birds and domestic poultry in this outbreak. In both cases, the initial virus detection occurred in the laboratory of the Scientific Veterinary Institute “Novi Sad” and the confirmatory diagnosis was provided by the National Reference Laboratory for Avian Influenza in the Specialized Veterinary Institute “Kraljevo”, Serbia. After laboratory confirmation of HPAI virus, the Serbian government implemented control measures through the Veterinary Directorate of the Ministry of Agriculture, Forestry and Water Management, with the aim to stop the spread of the virus and reduce the risk of losses in the poultry industry.

This is the first official and comprehensive study of the epidemiology and pathology of highly pathogenic avian influenza virus in poultry on the territory of our country ever documented. This research was carried out with the aim of describing the first H5N8 outbreak in backyard chickens in Serbia, and it includes the clinical signs, mortality, gross and microscopic lesions, immunohistochemistry (IHC), real-time reverse transcriptase (rRT) PCR and virus characterization. Additionally, this study documents the laboratory identification and pathology associated with the HPAI H5N1 in non-commercial poultry in Serbia for the first time, with successful suppression of the disease in both cases.

## 2. Materials and Methods

### 2.1. Sampling

In February 2017, 15 fresh chicken carcasses were collected and submitted to Scientific Veterinary Institute “Novi Sad” in order to determine the cause of death, due to an acute increase in flock mortality. These chickens were a domestic hybrid breed, about 40 weeks old, collected from a backyard flock in the Central Banat district, in the Province of Vojvodina. The backyard where the infection appeared is located in a rural, low-traffic region, close to a large standing water surface—the Danube-Tisa-Danube canal. In addition, a fish pond with a large number of wild birds was located approximately five kilometers away from this backyard holding. The flock consisted of 46 birds (both hens and roosters). In May 2022, several cases of sudden death of domestic poultry were detected in three isolated holdings that were in direct contact (contact holdings) in a village in the Central Banat district, in the Province of Vojvodina. The holdings were located in close proximity to three large ponds and the Tamish River. In all three households, there were no more than 15 birds in a flock, and they were mainly hens and roosters. The samples of 10 dead birds were sent to the Scientific Veterinary Institute “Novi Sad” for pathomorphological and virological diagnostics due to suspicion of HPAI. In both cases, the animals were raised in an extensive system—on a wide backyard area where the birds could move freely, without implementation of any biosecurity measures, which is the most common method of poultry breeding in the villages in the Province of Vojvodina. No mortality and morbidity were observed in other backyard holdings in the same or surrounding villages during the two epidemics.

The postmortem examinations and tissue sampling were performed according to a standard protocol; external body and internal organs were examined grossly, and the gross pathology was recorded and photographed. Tissue samples of the brain, lung, heart, pancreas, intestine, kidney and spleen were used for histopathological examination. To study the distribution of avian influenza virus (AIV) antigen in various tissues, IHC was performed on most organs taken from chickens that died during epizootic in 2017. Brain, spleen, lung and pancreas tissue samples were used for molecular diagnosis by RT-PCR. Additionally, oropharyngeal swabs were taken from two to five clinically ill chickens from each flock for PCR testing. After the detection of avian influenza virus in the samples, as it was (in this case) a priority disease in detection procedure, the examinations for other viruses and bacteria were not carried out.

### 2.2. Histopathology and Immunohistochemistry

After fixation in 10% buffered formaldehyde for 48 h, the tissue samples were dehydrated through graded series of alcohol and embedded in paraffin. The sections were cut at 4–5 μm and stained with hematoxylin and eosin. Duplicate sections were immunohistochemically analyzed using a commercial Novolink Polymer Detection Systems staining kit, Novocastra (Leica biosystems, Nussloch GmbH, Germany). For immunohistochemical demonstration of viral antigen, selected paraffin-embedded sections were deparaffinized and rehydrated in xylene and a graded series of alcohol. Citric buffer was incubated in a microwave oven (560 W) for 21 min for antigen retrieval. To detect the influenza A virus antigen, the sections were incubated with rabbit antinucleoprotein serum in a dilution of 1:1000 for 1 h in a humid chamber at room temperature, as described previously [17]. The immunoreaction was visualized using diamino benzidine (DAB) solution and counterstained with Mayer’s hematoxylin. A semiquantitative scoring system (negative, mild, moderate, and marked) was used to evaluate the intensity and extension of the influenza A virus antigen immunostaining of different tissues. Appropriate positive and negative control sections were included. Additionally, the selected brain tissue sections were immunohistochemically examined for the presence of astrocytes expressing glial fibrillary acidic protein. These sections were immunostained with a specific marker (GFAP, Dako, Glostrup, Denmark) applied at 1:400 dilution. Endogenous peroxidase activity was blocked with peroxidase block for 20 min and antigen retrieval was accomplished by treating the sections with proteinase K (Dako, Glostrup, Denmark) for 6 min at room temperature.

### 2.3. RT-PCR Assay

The presence of the H5N8 HPAI virus in collected samples from season 2016/2017 and presence of the H5N1 HPAI in collected samples from May 2022 was performed in pooled tissue suspensions and selected oropharyngeal content in PBS by molecular diagnostic methods. Tissue samples (brain, spleen, lung) of dead chickens were cut in small pieces weighing 0.2 g and placed in 2 mL micro tubes and homogenized in 1 mL sterile phosphate-buffered saline for 5 min using a TissueLyser LT (Qiagen, Hilden, Germany) operating at 50 Hz. The homogenates were then centrifuged for 10 min at 2000× *g*. Supernatant was used for RNA extraction. Oropharyngeal swabs from animals showing clinical signs of the disease were immersed in 0.75 mL of sterile phosphate—buffered saline during the sampling, placed on ice packs during the transport to the lab, and after lab reception, vortexed vigorously for 3 min. The supernatants were directly used for RNA extraction. Total RNA was extracted using the commercial IndiSpin Pathogen Kit (Indical Bioscience GmbH, Leipzig, Germany) according to the manufacturer’s instructions.

The presence of avian influenza virus genome RNA was done by TaqMan-based one-step reverse transcription real-time PCR (RT-qPCR). The detection of matrix gen (M gen), suitable for detection of all influenza viruses, and detection of H5 gene was performed by a procedure using oligonucleotide primers and probes and thermal profiles described by Spackman et al. [19], and by using commercial kit RNA UltraSense™ One-Step Quanti-tative RT-PCR System (Invitrogen, ThermoFisher Scientific, Waltham, MA, USA), according to the manufacturer’s instructions. Both the N8 and N1 genes were detected by TaqMan-based RT-qPCR technique with oligonucleotide primers and probe and thermal profile as described by Hoffman et al. [20], and by using the same, previously described, commercial one-step RT-qPCR kit. A highly pathogenic pathotype was confirmed by Sanger sequencing of the hemagglutinin (HA) gene, as described by Slomka et al. [21].

For HA gene cleavage site sequencing, primers and conditions described by Slomka et al. [22] were used. PCR products were purified using mi-Gel Extraction kit (Metabion, Germany), according to the manufacturer’s protocol. Purified PCR products were directly sequenced on both strands on Genetic Analyser 3130 (Applied Biosystems, Carlsbad, CA, USA) using BigDye 3.1 sequencing kit (Applied Biosystems, Carlsbad, CA, USA) and with the same primers that were used for the PCR. The obtained sequences were compared in GenBank (National Centre for Biotechnology Information, Rockville, MD, USA) using the nucleotide Basic Local Alignment Search Tool program (https://blast.ncbi.nlm.nih.gov/Blast.cgi (accessed on 10 February 2023).

## 3. Results

### 3.1. Gross Pathologic Lesions

Clinically, what both cases have in common is that most birds had ruffled feathers and a majority of them had nervous manifestations (such as ataxia and leg paralysis) (Figure 1a), later accompanied by 100% mortality in the flock. 

External examination of chicken carcasses showed mild (H5N8-infected chickens) to moderate (H5N1-infected chickens) cyanosis and necrosis in the crest and wattle in 10 examined birds (Figure 1b). No other skin or subcutaneous lesions were detected in the remaining chickens. All examined chickens were in good body condition, with sufficient body fat content. The necropsy examination revealed severe and widespread hemorrhagic and necrotic lesions in internal organs, including serosal and mucosal surfaces. Focal hemorrhages in pectoral muscles were found in six chickens. A consistent finding in all birds was petechiae in coelomic fat (Figure 2a,b). All infected chickens showed subepicardial petechial hemorrhages (Figure 3a) and petechiae in the epicardial fat (Figure 3b), while ecchymoses of various intensities in the endocard were occasionally seen. In many cases, the spleen was enlarged and multiple pin-point whitish necrotic foci were present. Intestinal mucosa was diffusely hyperemic in a few birds: hemorrhages in the mucosa and bloody mucinous exudate in the lumen of the small intestine were detected in 10 chickens. The lungs were dark red and the lung lesions were characterized by massive hemorrhages, intensive edema and congestion in most of the examined birds (Figure 4a). Air sacs were thickened in a few birds. The ovarian follicles were hemorrhagic and severely congested (Figure 4b). Additionally, the kidneys were congested and swollen. The livers were slightly edematous and fragile with petechial hemorrhages, but this was not a consistent finding. Hyperemia of the meningeal and cerebral blood vessels was also noticed. In both cases, there were no significant macroscopic lesions in the proventriculus and gizzard. Unlike H5N8-infected chickens, with no macroscopically visible changes in the pancreas, macroscopic hemorrhagic-necrotic changes were observed in H5N1-infected chickens (Figure 5a,b).

### 3.2. Microscopic Lesions

Microscopic lesions were present in many organs. Microscopically, the most severely affected organs were the lungs, heart, pancreas, kidneys, spleen and brain, while only the intestines did not show significant microscopic changes. Lesions in these organs were mainly characterized by necrosis with mononuclear cell infiltrates as well as hemorrhages. In the lungs of all chickens, severe congestion, hemorrhages, and blood extravasations from the ruptured vascular wall were found (Figure 6a). In the heart, subepicardial hemorrhages accompanied with hemosiderosis were seen, while myocardial degeneration was present in two chickens (Figure 6b). The spleen was characterized by severe congestion, hemorrhage and diffuse necrosis. In kidneys, hemorrhages, focal degeneration and necrosis of the renal tubular epithelial cells were observed. Although there were no macroscopic lesions in the pancreas, multifocal necrotic foci with mild to moderate mononuclear cell infiltrations were found (Figure 7a). In the exocrine pancreatic cells, vacuolization of the cytoplasm was detected. In the brain, lesions consisted of disseminated focal microgliosis, necrosis and infiltration of mononuclear cells around blood vessels. Non-suppurative meningitis, neuronophagia and encephalomalacia were observed in ten cases (Figure 7b). Mild lymphocytic infiltration in the lamina propria was present in the intestines. The microscopic changes observed were the same in terms of frequency and intensity in both examined groups.

### 3.3. Immunohistochemistry

#### 3.3.1. Influenza Viral Antigen

Influenza viral nucleoprotein was detected by immunohistochemistry in the following organs: brain, lung, heart, pancreas, intestine, kidney and spleen. In general, there was a strong parity between the demonstration of viral antigen and the identification of histologic lesions (Table 1). The antigenic staining was nuclear and cytoplasmic in distribution. A positive reaction was detected in all the examined chickens. In the brain, intense nuclear and cytoplasmic positive staining was detected focally, and a varying number of positive cells ranged from a few positive neurons to a vast number of positive cells in some brain areas. Influenza viral nucleoprotein immunoreactivity was observed in glial cells in the molecular and granular layer (Figure 8a). Viral antigen (both nuclear and cytoplasmatic) was also detected in endothelial cells of the brain blood vessels (Figure 8b). Additionally, intense expression of viral antigen was detected in the cytoplasm of necrotic acinar cells in the pancreas. In the lungs, intense antigenic staining was detected in scattered macrophages, epithelial and blood endothelial cells (Figure 9a). Tissue macrophages and endothelial cells of the spleen were immunopositive (Figure 9b). In the heart, an intense reaction was detected in scattered cardiac myocytes (Figure 10a). In the kidney, viral antigen was detected in the tubular epithelium and endothelium of the glomerular capillaries (Figure 10b). In four chickens, only a few immunopositive cells (enterocytes) in lamina propria were detected in the intestines. No nucleoprotein antigen was detected in negative controls. The immunohistochemical results of the cases are shown in Table 1.

#### 3.3.2. Astrocytes

The astroglial reaction showed numerous Glial Fibrillary Acidic Protein (GFAP)—positive astrocytes in the cerebrum and cerebellum in all cases of the studied brain sections. The distribution of stained astrocytes was prominent in the areas of an intense inflammatory response (Figure 11a,b). The presence of GFAP-positive cells in the brain of the negative controls was within physiological limits. 

### 3.4. Molecular Findings 

The presence of matrix gene of influenza viruses were detected in tissue/swab samples of all tested animals by RT-qPCR method. The virus load in the samples was very high, and the Ct values were between 13 and 20. In addition, the presence of H5 and N8 gene (H5N8 virus subtype) of influenza virus was confirmed in all influenza virus positive poultry samples collected in February 2017 by subtyping. However, in all influenza virus positive poultry samples collected in infected backyards in May 2022, the presence of H5 and N1 gene (H5N1 virus subtype) of influenza virus was confirmed by the same methodology. Sanger sequencing of the hemagglutinin (HA) gene confirmed a highly pathogenic influenza virus pathotype (HPAI). The H5N8 strain identified in this study clustered in clade 2.3.4.4b has a close genetic relationship with isolated HPAI H5N8 strains from Belgium, Croatia, Hungary and France. The detected H5N1 strain from 2022 identified in this study that clustered in the same clade, has a highest genetic similarity of hemagglutinin gene (99%) with isolated HPAI H5N1 strains detected in 2021 from turkey in Poland, chicken from The Netherlands, mute swan from Croatia, duck from Israel and goose from the Czech Republic.

## 4. Discussions

Until the worldwide massive HPAI H5N1 outbreaks in 2005–2006, HPAI viruses have rarely induced high mortality rates in wild bird species. However, since these outbreaks, HPAI H5N1 has continued to cause illness and death in a variety of wild birds in Asia as well as in Europe, and mute swans were one of the most frequently affected wild bird species [23,24,25]. Some authors characterized them as an indicator species for the avian influenza virus [26,27] and pointed out their essential role in virus transmission in Europe. Over the last two decades, three HPAI outbreaks were recorded in the Republic of Serbia—H5N1 in 2006, H5N8 in 2016/2017, and H5N1 in 2021/2022. The main similarities between these outbreaks were reflected in a small number of affected poultry, and mute swans were the most commonly affected wild bird species, with a high mortality rate. Present findings also suggest that mute swans certainly had a significant role in spreading the disease in the affected areas. However, these birds are mostly sedentary and partially migratory, so therefore, it could be assumed that they cannot spread the virus over long distances. The HPAI H5N8 and H5N1 viruses were most likely introduced into the country through migratory wild birds, as viruses of this subtype have been previously reported in our neighboring countries Hungary, Croatia and Bulgaria, as well as in other European countries [28,29,30,31]. It is important to point out the HPAI epidemic that occurred in the 2021/2022 epidemiological year was the most widespread ever in Europe and HPAI H5N1 was by far the predominant virus type reported [31]. However, a significantly lower number of infected birds, both wild and poultry, were recorded in Serbia compared to the 2016/2017 epidemiological year.

This review describes the first HPAI H5N8 and H5N1 infection in backyard poultry in our country. During both epidemics, a small number of outbreaks were recorded in domestic poultry (4 in 2016/2017 and 3 in 2021/2022), with low mortality and morbidity rate, which is not expected for HPAI virus that normally has high mortality rates on the field [32]. What all these outbreaks had in common was that the poultry were raised extensively, free to move in the backyard, meaning they could have contact with wild birds. This method of poultry breeding is typical for the Vojvodina region, where all cases of poultry infection have been recorded. Immediately after the disease was confirmed in these backyard holdings, strict measures were applied to prevent the spread of the disease, through thorough cleaning, disinfection of facilities, and testing of poultry from contact holdings where no new cases of disease or death were recorded. An “infected zone” with a radius of 0–3 km was established around the infected yard, while a “threat zone” was imposed measuring between 0 and 10 km in radius from the infected zone. The control measures included implementation of active and passive surveillance—i.e., control of poultry health in potentially risky places near water areas with wild birds, confinement of poultry in facilities, prevention of direct and indirect contacts between wild birds and poultry, and raising biosecurity and hygiene measures on farms in order to prevent the occurrence and spread of the disease. In both cases, the slow spread of the virus throughout the country possibly indicates that measures implemented to control the disease were effective, and consequently, no mass distribution or dissemination of the disease was recorded.

The current study did not investigate other viral or bacterial agents other than HPAI virus. However, epidemiological data, clinical signs, molecular findings and pathological lesions clearly indicate HPAI virus infection in domestic chickens. The virus was confirmed by real-time RT-PCR and sequencing of the HA gene. 

Natural infection by HPAI viruses results in various pathological outcomes that depend on virus strain, host species, and environmental factors. One of the typical external gross lesions in HPAI virus infections in gallinaceous species includes edema of the head and face and subcutaneous hemorrhage with cyanosis in the skin, particularly in the comb and wattle [33]. In both HPAI infections in chickens, these findings were seen in a few examined birds in the form of slight cyanosis of the comb and wattle. The necropsy of the chickens revealed severe hemorrhagic, necrotic and inflammatory changes in most organs. Grossly, the most affected organs were the lungs and heart in both cases. The predominant lesions were subepicardial hemorrhages in the heart, and congestion, hemorrhages and edema of the lungs. Previously, the same gross lesions were reported in HPAI H5N1-infected domestic chickens and other gallinaceous species [34,35]. as well as in H5N8-infected wild birds [17,36]. Although pancreatic lesions are common for HPAI in gallinaceous species [33], these gross lesions were not found in H5N8-infected chickens. However, microscopically, pancreatic damage associated with virus replication was observed. The significant microscopic pancreatic lesions were multifocal necrosis of the acini, and in the exocrine pancreatic cells in both cases. Vacuolization of the cytoplasm was detected, which is in agreement with Lean et al., Anis et al., and Núñez et al. [37,38,39]. Except for the pancreas, histopathological examination of the tissues collected post-mortem correlated with the gross changes. The predominant histologic lesions were found in the brain, pancreas and lungs, manifested as nonsuppurative encephalitis in the cerebrum, multifocal necrosis of the acini, and severe congestion and hemorrhage, respectively. In the heart, myocardial degeneration was described, and previously similar lesions were reported in H5N8-infected mule ducks [30]. The microscopic findings in the lungs are consistent with those observed in the natural infection of domestic waterfowl [38].

Immunohistochemical detection of the influenza virus in all examined organs revealed organotropism of the H5N8 virus in domestic chickens. Positive immunohistochemical signal for the virus was detected in all chickens (15/15), and viral antigen was found in multiple tissues, confirming the pantropic nature of this HPAI virus in this poultry species. A similar immunoreaction pattern was seen in layers and turkeys naturally infected with HPAI H5N8 [40]. Distribution of viral antigen was mainly associated with microscopic lesions. In this study, vascular endothelial cells were positive in various tissues. It is known that HPAI viruses replicate in vascular endothelial cells, which are very important for the dissemination and systemic infection of the HPAI viruses [33,41]. According to previous research, HPAI H5N8 exhibits immunopositive reactivity in vascular endothelial cells in different domestic bird species, including ducks [30,39], chickens and turkeys [40], and some wild bird species [42], although this is not always a consistent finding [43]. Chickens infected with H5N8 virus abundantly expressed influenza virus antigen in respiratory epithelium, which would support a respiratory or airborne infection route. In addition, different studies show that respiratory tract infection is considered the main source of HPAI virus excretion from the oropharynx in wild birds [44]. In contrast, the virus antigen expression in the intestinal epithelium was more limited, showing a low level of attachment to the intestinal epithelium. Although enterotropism is more commonly seen for LPAIVs, recently reported studies for HPAI H5N8 in some wild birds showed fecal-oral virus transmission [42].

The novel finding of this study was the astroglial reaction in the form of hyperplasia and hypertrophy in brain sections in the majority of H5N8-infected chickens. In brain sections of infected birds, GFAP-positive astrocytes were detected in high numbers. It has been previously described that astrocytosis and astrogliosis are present in other viral bird encephalitis infections [45]. However, in some wild birds infected with HPAI H5N1‚ there was no reaction of astrocytes using GFAP [46]. Various studies have proven that the proliferation of astrocytes after a CNS injury requires several days, which is also influenced by the type of CNS damage and the individual’s age [47]. In this study, astroglial reaction and IHC-positive neurons indicate a high neurotropism of the HPAI H5N8 virus. These findings of the neurotropic nature of the H5N8 virus correspond to the results of other researchers on HPAI H5N8 infections in domestic poultry [38], Pekin ducks [29], mule ducks [30] and wild waterfowl species [18,36], as well as in birds naturally infected with H5N1 [46]. 

## 5. Conclusions

In conclusion, the present study provides a pathological and clinical comparison of chickens naturally infected with HPAI H5N8 involved in an HPAI epizootic registered in Serbia in winter 2016/2017 and HPAI H5N1 registered during 2021/2022. This study showed that in both epidemics, viral infection in chickens resulted in hemorrhagic and necrotic lesions in systemic organs, including the lungs, heart, brain, pancreas, and kidneys, leading to multiorgan failure. Gross pathological findings were a critical factor in the detection of the infection, as the clinical and pathologic presentation of this case was strongly suggestive of HPAI. Following these outbreak cases, strict biosecurity measures were implemented in the control endangered areas, and no further positive cases in poultry were detected during subsequent surveillance and AIV screening. These outbreaks highlight the importance of rapid detection, rapid notification, and a response system in order to stop the spread of the disease.

## Figures and Tables

**Figure 1 animals-13-00700-f001:**
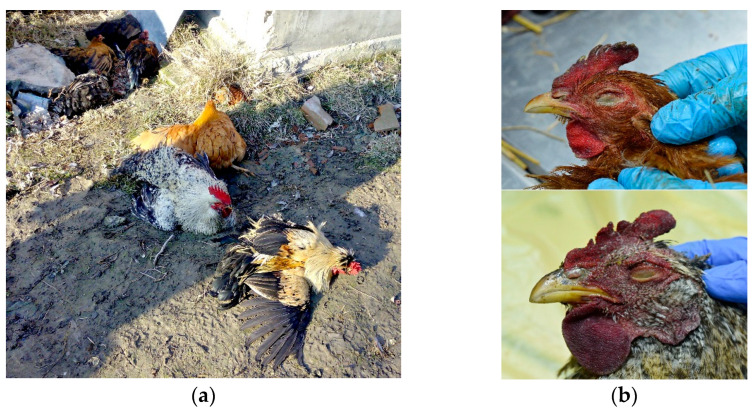
(**a**) Clinical signs of H5N8 avian influenza virus infection in chickens—ataxia and leg paralysis. (**b**) Cyanosis of comb and wattle in H5N8-infected chicken (upper photo) and H5N1-infected chicken (lower photo).

**Figure 2 animals-13-00700-f002:**
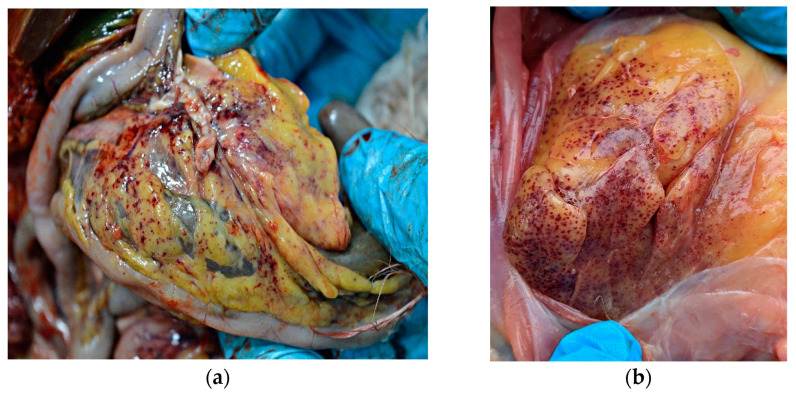
Gross lesions in chickens naturally infected with HPAI virus. (**a**) Petechiae in coelomic fat (H5N8-infected chicken). (**b**) Diffuse petechial hemorrhage in coelomic fat (H5N1-infected chicken).

**Figure 3 animals-13-00700-f003:**
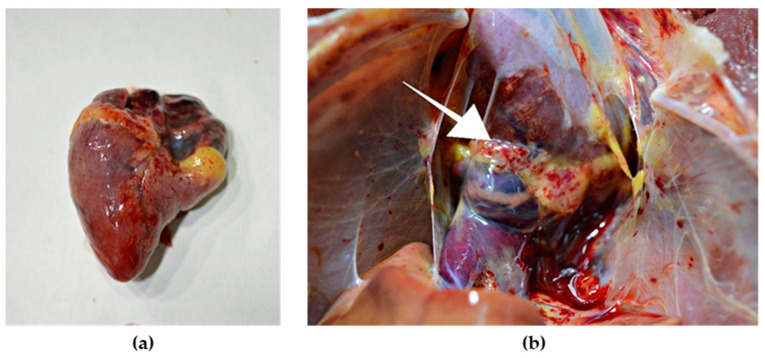
Gross lesions in the heart of chicken naturally infected with HPAI H5N8. (**a**) Petechial hemorrhage in the apical zone and base of the heart. (**b**) Petechial hemorrhage in the epicardial fat (arrow).

**Figure 4 animals-13-00700-f004:**
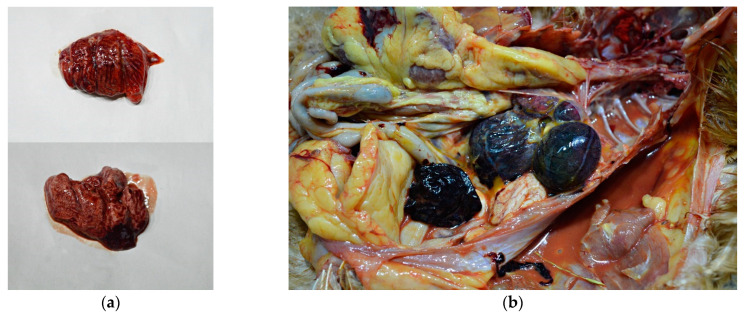
Gross lesions in chickens naturally infected with HPAI H5N8. (**a**) Extensive congestion and hemorrhages in the pulmonary parenchyma. (**b**) Severely congested ovarian follicles.

**Figure 5 animals-13-00700-f005:**
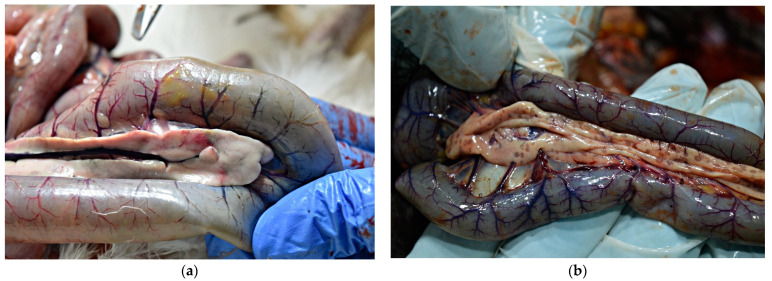
Gross lesions in chickens naturally infected with HPAI H5N1. (**a**) Pancreas, multifocal, partially coalescing hemorrhages. (**b**) Multiple necrotic foci 0.1 to 0.5 mm in diameter in the pancreas.

**Figure 6 animals-13-00700-f006:**
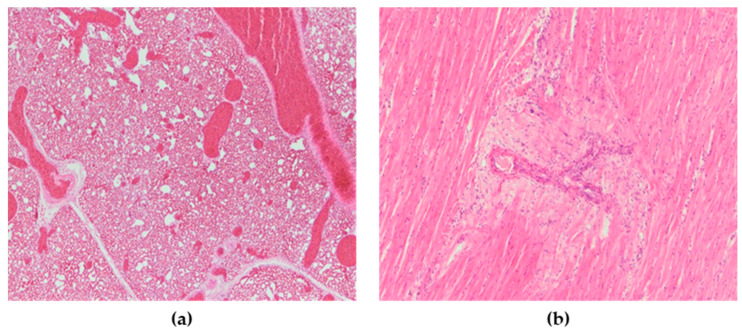
(**a**) Lungs, severe vascular congestion. H&E × 100. (**b**) Heart, myocardial degeneration, H&E × 200.

**Figure 7 animals-13-00700-f007:**
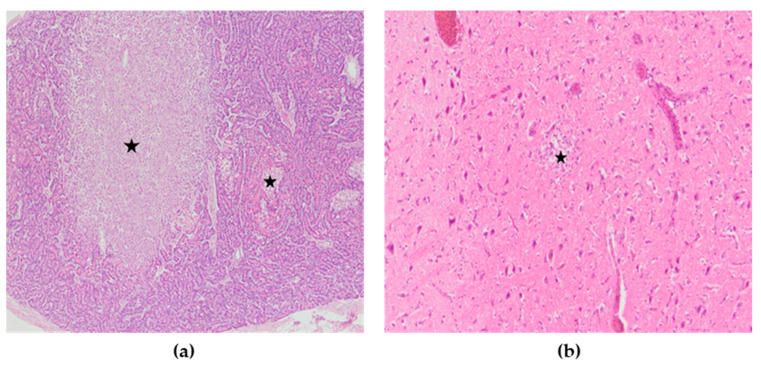
(**a**) Pancreas, multifocal areas of necrosis of pancreatic acini (asterisk), H&E × 100. (**b**) Cerebrum, encephalomalacia (asterisk), H&E × 200.

**Figure 8 animals-13-00700-f008:**
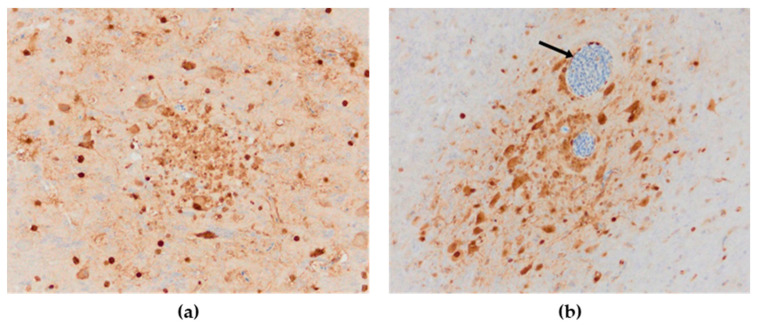
Immunohistochemical distribution of influenza viral nucleoprotein antigen in the tissues of chickens infected with H5N8 HPAI. (**a**) Brain. Brown positive staining of the glial cells associated with areas of encephalomalacia, bar × 400. (**b**) Brain. A large amount of AIV nucleoprotein in the nucleus and cytoplasm of neurons and glial cells in the cerebrum; immunopositive endothelial cells (arrow), bar × 400.

**Figure 9 animals-13-00700-f009:**
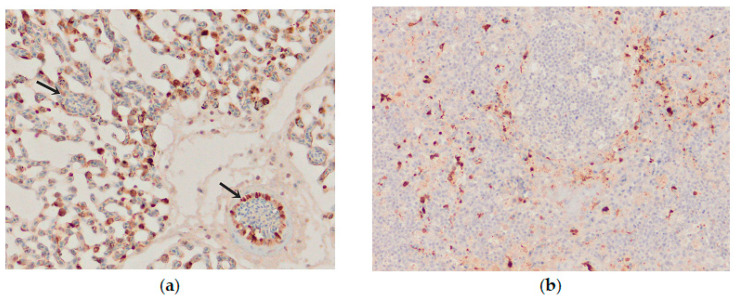
Immunohistochemical distribution of influenza viral nucleoprotein antigen in the tissues of chickens infected with H5N8 HPAI. (**a**) Lungs. Macrophages and epithelial cells of the lung interstitium, and vascular endothelial cells (arrows) positive for AIV antigen, bar × 400. (**b**) Spleen, AIV nucleoprotein in the vascular endothelial and inflammatory cells, bar × 100.

**Figure 10 animals-13-00700-f010:**
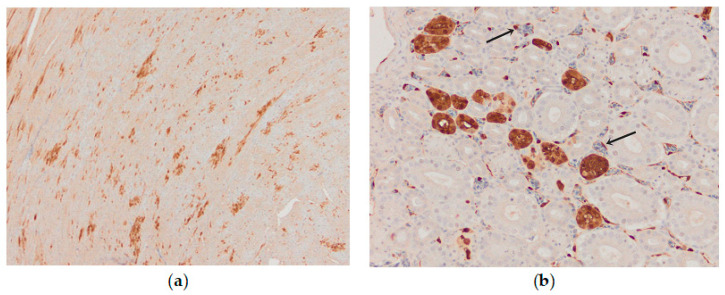
Immunohistochemical distribution of influenza viral nucleoprotein antigen in the tissues of chickens infected with H5N8 HPAI. (**a**) Heart, AIV nucleoprotein in the nucleus and cytoplasm of myocytes, bar × 100. (**b**) Kidney, AIV nucleoprotein in the tubular epithelial cells and vascular endothelial cells (arrow), bar × 400.

**Figure 11 animals-13-00700-f011:**
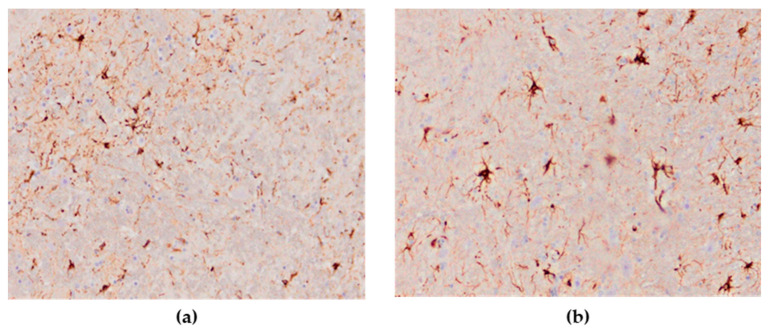
Numerous GFAP-positive astrocytes in the brain of HPAI H5N8 positive chickens (original magnification (**a**) × 100, (**b**) × 400).

**Table 1 animals-13-00700-t001:** Histopathological changes and average distribution of viral antigen–positive cells in tissues sampled from domestic chickens naturally infected with the HPAI H5N8 virus.

Organ	Histopathology	IHC Score	IHC Positive Cell Type
Brain	Microgliosis and necrosis, non-purulent meningitis, neuronophagia and encephalomalacia	+++	Neurons, glial cells, endothelial cells
Pancreas	Necrosis	+++	Necrotic acinar epithelial cells
Lung	Congestion, hemorrhages	+++	Endothelial cells, macrophages
Heart	Subepicardial hemorrhages, myocardial degeneration	++	Cardiac myocytes
Intestine	Catarrhal enteritis	+	Enterocytes
Kidney	Tubular necrosis, hemorrhages	++	Tubular epithelial cells; endothelium of the glomerular capillaries
Spleen	Necrosis, hemorrhages	+++	Endothelial cells, macrophages

Viral antigen: +++ widespread (marked); ++ multifocal (moderate); + infrequent (mild).

## Data Availability

Not applicable.

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
