# Peer review of "Highly Pathogenic Avian Influenza H5N8 Outbreak in Backyard Chickens in Serbia"

_animals, 2023, doi:10.3390/ani13040700_

Round 1

Reviewer 1 Report

The submitted manuscript is a simple case report of highly pathogenic avian influenza (HPAI) outbreak in backyard flock that occurred almost 6 years ago in Serbia. The article does not provide any novel data about pathology of HPAI and it does not merit publication in the journal.

Reviewer 2 Report

The authors should kindly consider the comments in the attached file

Reviewer 3 Report

Study of an Outbreak of HPAI H5N8 in Backyard Chickens in Serbia.

This manuscript by Biljana Djurdjevic et al is well written  and I have a few comments that need to be address by authors:

Line 222, 223 and 225—replace Histopathologic with microscopic

Line 242—I disagree with the authors that the photomicrograph 5a shows hemorrhage.  It appears to be severe congestion of vessels to me.

Line 243. Figure 5b—I can’t see lymphocytes in the heart at the magnification given.

Line 261. Alveolar macrophages is an incorrect term since birds do NOT have alveoli.

Figure 8. Can you identify the mononuclear cells—are they macrophages, lymphocytes or plasma cells?  The magnification presented does not allow to determine cell types in 8a.  8b has poor contrast and can’t see staining very well at all.

Figure 9b doesn’t look like renal parenchyma unless the majority of the tubules are missing.

Lines 354 and 363 replace histopathologic with microscopic

Round 2

Reviewer 1 Report

I maintain my stance that this article should not be published in an international journal.

Any scientific publication should present the results that add to existing knowledge and in this case the novelty of the findings is close to zero. The authors might consider publishing the manuscript in one of the local journals.